# The Impacts of Traditional Ecological Knowledge towards Indigenous Peoples: A Systematic Literature Review

**Jamilah Mohd Salim** [1,2], **Siti Nursyadiq Anuar** [1], **Khatijah Omar** [1,3], **Tengku Rozaina Tengku Mohamad** [1,4] **and Nur Azura Sanusi** [3,5,*]

1 Institute for Tropical Biodiversity and Sustainable Development, Universiti Malaysia Terengganu, Kuala Nerus 21030, Terengganu, Malaysia
2 Faculty of Science and Marine Environment, Universiti Malaysia Terengganu, Kuala Nerus 21030, Terengganu, Malaysia
3 Faculty of Business, Economics and Social Development, Universiti Malaysia Terengganu, Kuala Nerus 21030, Terengganu, Malaysia
4 Faculty of Fisheries and Food Science, Universiti Malaysia Terengganu, Kuala Nerus 21030, Terengganu, Malaysia
5 Higher Institution Centre of Excellence, Institute of Tropical Aquaculture and Fisheries, University Malaysia Terengganu, Kuala Nerus 21030, Terengganu, Malaysia
* Correspondence: nurazura@umt.edu.my

**Abstract:** Indigenous peoples are groups with different cultural and social characteristics that share inherited ties to their homeland and natural resources. They have their own understanding and cultural experience that amounts to traditional ecological knowledge. The aim of this study is to identify the impacts of traditional ecological knowledge on indigenous people. Two main databases, namely Web of Science and Scopus, were used to conduct a systematic literature review. From the findings and analysis, two themes and eleven sub-themes were identified. The first theme is economic activities, including six sub-themes: sources of income, employment opportunities, offering products to vendors or buyers, providing market value, providing low treatment cost, and providing opportunities to develop micro-enterprises. The second theme is health, with five sub-themes: supporting food security, harvesting country food, food or plant benefits, perceived health or medicinal purposes, and livelihoods of the indigenous people. In conclusion, traditional knowledge can play an important role in contributing to the livelihoods of indigenous people. In general, traditional knowledge can help indigenous people to improve their quality of life, especially those who rely on natural resources to survive, by offering secure and supplemented food, for instance, as well as a source of earnings, crucial for food security during hard times. Additionally, traditional knowledge of wild edible and medicinal plants can play a significant role in a community's capacity to remain resilient and be preserved for future generations.

**Keywords:** traditional ecological knowledge; indigenous people; economics; health; systematic literature review

## 1. Introduction

A formal definition of "indigenous" has not been offered by any UN organization due to the multiplicity of indigenous peoples. However, the system has produced a modern interpretation of the term based on Jose R. Martinez Cobo's study, wherein "indigenous people" are defined as follows: "Indigenous communities, peoples, and countries are those that perceive themselves as distinctive from other groups of the civilizations that currently rule those territories or parts of them because of their heritage with pre-invasion and pre-colonial civilization that originated on their territory. In accordance with their own cultural patterns, social institutions, and legal system, they are currently non-dominant sectors of society who are utterly convinced about conserving, developing, and passing

on to subsequent generations their ancestral lands and cultural beliefs as the basis of their continued existence as peoples".

According to the World Bank [1], indigenous peoples are groups with different cultural and social characteristics that share inherited ties to their homeland and natural resources. They are completely reliant on the land and natural resources to live their daily lives, and they are inextricably linked to personalities, beliefs, and livelihoods. Furthermore, they maintain their own languages, which are different from those of the mainstream societies or communities. By adhering to unique traditions, they develop social, cultural, economic, and political traits that are unique compared to the dominant communities where they live. Currently, the estimated total population of indigenous people worldwide is 476 million. Even though they only make up approximately 6 percent of the world's population, they represent about 19 percent of the extremely poor [2].

Another definition of indigenous people is that they are a particular, distinct cultural group that uphold their own inherited cultural and social uniqueness, socioeconomic system, and administrative system [3,4]. Other than that, they reside in rural parts of the continent and depend solely on the natural resources available locally for their livelihood. Indigenous people have their own understanding and cultural experiences that are built into relationships between humans, non-humans, and other than humans in certain ecosystems. Figure 1 shows the names of tribes or communities in the areas of the selected studies.

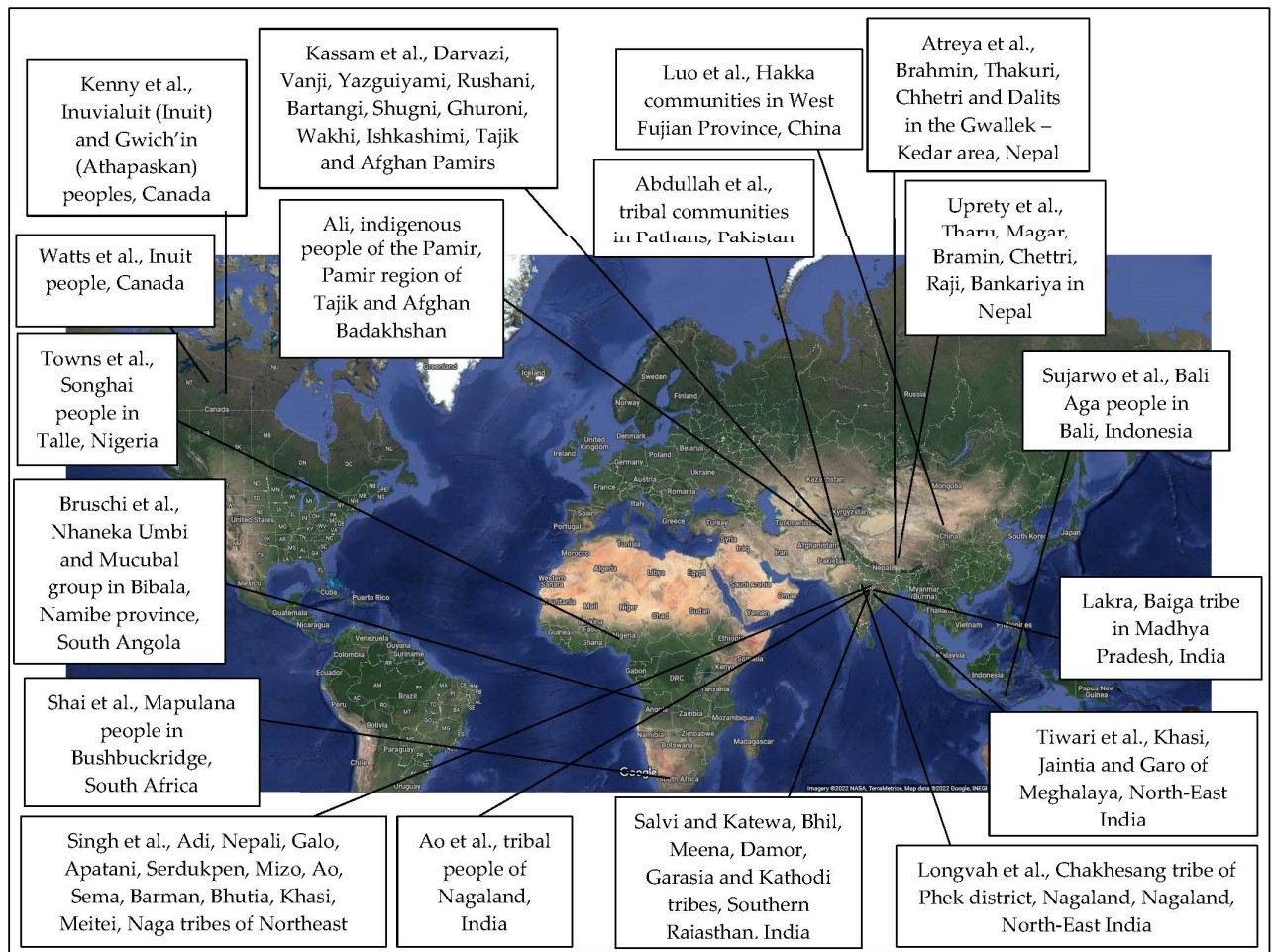

**Figure 1.** The names of tribes or communities in the areas of the selected studies (Source: https://www.google.com/maps/@42.9590663,12.3572352,13660864m/data=!3m1!1e3, accessed on 7 December 2022) [5–23].

As a global organization, the World Bank has adopted a variety of programs with indigenous communities, including increasing participation and awareness of indigenous peoples' rights in a number of countries. According to a World Bank news release from 2020, the World Bank has approved a funding initiative in the form of a loan amounting up to USD40 million for various groups in Ecuador, including indigenous, Afro-Ecuadorian, and Montubio peoples and nations. The financing was awarded for COVID-19 pandemic recovery operations in the areas of governance and economic development. In a similar way, the World Bank helped Vietnam become more involved in the economy by working with different sectors and putting extra focus on projects that help ethnic minorities, women, and other vulnerable groups obtain jobs and make money [24].

Traditional ecological knowledge is defined as a network of knowledge, beliefs, spirituality, and traditions aimed at maintaining and connecting indigenous relationships with heritage, culture, and place, which are transferred directly and indirectly among kin communities [25]. Meanwhile, Harun and Othman [26] elaborated that traditional ecological knowledge is also known as indigenous knowledge and is used by the local community to make a living in a particular environment. Traditional indigenous knowledge is thought to encompass knowledge, spirituality, and beliefs passed down from generation to generation via heritage and the cultural transmission of livelihood with each other and with the environment.

This research makes a contribution to the sustainable development goals by addressing the objectives of goal 1, which is to end poverty; goal 2, which aims to end world hunger; and goal 3, which aims to ensure that all people enjoy good health and wellbeing. According to statistics gathered by the World Bank in 2016 [27] from 90 different countries, indigenous peoples make up 15 percent of the world's population of people who live in extreme poverty, making them the poorest of the poor. As a direct consequence of this, the second goal is to increase the productivity of indigenous peoples while simultaneously doubling their incomes. Goal 3 places an emphasis on wellbeing by aiming to achieve universal health coverage, encompassing financial risk protection, access to quality essential health care services, and access to essential medicines and vaccines that are safe, effective, of high quality, and affordable for all people.

Indigenous peoples are not only the keepers of one-of-a-kind belief systems and knowledge systems, but they also possess invaluable knowledge regarding sustainable practices for the management of natural resources. This is because indigenous peoples have been living on their land for a long time. They have a special connection to the land that has been in their family for many years, and they make productive use of it. In terms of both their current level of physical wellbeing and the cultural customs they have passed down through the generations, the land that their ancestors once inhabited is of the utmost significance to their continued existence as a distinct nation. Indigenous peoples have their own distinct concept of development that is based on the values, visions, needs, and priorities that have been ingrained in their culture for generations. As a result, it is critical to ensure that their traditional knowledge is preserved for future generations, particularly when it comes to the use of various medicinal plants.

The awareness that traditional ecological knowledge might contribute to the study of biodiversity is a main reason why interest in this knowledge has been increasing in recent years [5,28–30], as has interest in climate change [31–34], food security [32,35,36], medicine [6,37,38], and agriculture [39,40]. There are different scholars who share an interest in traditional or local knowledge for scientific [41–43], social [44,45], or economic reasons [46–48]. Hence, this study will employ a systematic literature review (SLR) to identify the impacts of traditional ecological knowledge on indigenous people. The process of the systematic review is based on the following research questions:

1. What are the types of traditional ecological knowledge (TEK)?
2. What are the impacts of TEK on economic activities?
3. What are the impacts of TEK on human health?

## 2. Methodology

Mengist et al. [49] underlined that an SLR differs from traditional narrative reviews by using reproducible, transparent, and scientific processes that allow for the collection of data and information that meet the pre-defined inclusion criteria. Researchers, scholars, and practitioners have used SLRs because they can help them produce more specific and accurate estimations on the issue of research [50]. Furthermore, an SLR is defined as a systematic and reproducible approach to locating, assessing, and synthesizing the existing results of completed or recorded investigations [51]. Before the systematic review process starts, the criteria are clearly laid out in an established protocol or strategy.

This section describes the method that was applied to gather the selected articles on the impacts of traditional ecological knowledge on economic activities and human health. The method employed in the study is PRISMA (Preferred Reporting Items for Systematic Reviews and Meta-Analyses); this method is described, followed by the resources, the screening process, eligibility criteria, the quality assessment, data abstraction, and analysis.

### 2.1. PRISMA

This study was guided by PRISMA, a published standard for an SLR. Commonly, the researcher refers to a published standard for guidance so that they can assess and implement the quality and rigor of the review [52]. PRISMA can also be used as the foundation for conducting a systematic review on other topics in the review report [53]. Sierra-Correa and Cantera Kintz [54] stated that PRISMA can outline the research questions based on a systematic review and is capable of classifying the inclusion and exclusion criteria for the study.

### 2.2. Database and Search Strategy

In the current study, two major journals indexed databases, Scopus and Web of Science (WOS), were identified for the review. According to Impellizzeri and Bizzini [55], there are numerous databases available, including SCOPUS, Google Scholar, EMBASE, MEDLINE (via PubMed), Web of Science, CINAHL, and Cochrane Controlled Trials; however, the study chose to focus on Scopus and Web of Science rather than other databases because their journal articles are confirmed to be indexed. Furthermore, these databases are reliable, offering empirical data and covering multidisciplinary study fields such as environmental sciences, agricultural sciences, biological sciences, and social sciences. According to Zhu and Liu [56], Scopus and WOS are the world's top and most competitive citation databases. Scopus is considered a new database in comparison to WOS, yet it can still compete with WOS. For example, Chinese researchers often use both databases to perform meta-analyses about their topics or investigations.

According to Shaffril et al. [57], identification, screening, and eligibility processes were applied to generate suitable articles for the study. These processes allowed the broad retrieval of articles and synthesis of the studies in order to provide a transparent and well-organized review. Keywords were identified in this study, namely, ecological knowledge, health, and food. To enrich these keywords, synonyms and related terms refer to the keywords used in previous studies. In the study, the combined keywords were put into the search string using truncation, the Boolean operators AND and OR, quotation marks, and asterisks (see Table 1).

**Table 1.** The keywords searching in the search string.

| Databases | Keywords |
|---|---|
| Web of Science | "traditional knowledg*" OR "ecological knowledg*" OR "indigenous knowledg*" OR "local knowledg*" OR "traditional ecological knowledg*" OR "local ecological knowledg*" (All Fields) and "human health*" OR "health*" OR "communit* health*" OR "indigenous health*" OR "health valu*" OR "public health" OR "global health" (All Fields) and "food safety" OR "food security" (All Fields) |
| Scopus | (TITLE-ABS-KEY ("traditional knowledg*" OR "ecological knowledg*" OR "indigenous knowledg*" OR "local knowledg*" OR "traditional ecological knowledg*" OR "local ecological knowledg*") AND TITLE-ABS-KEY ("human health*" OR "health*" OR "communit* health*" OR "indigenous health*" OR "health valu*" OR "public health" OR "global health") AND TITLE-ABS-KEY ("food safety" OR "food security")) |

*2.3. Screening Eligibility and Criteria*

Screening is the process of inserting or eliminating articles based on determined criteria. In order to produce appropriate articles in the SLR process, the screening process includes determining eligibility, inclusion, and exclusion criteria. The first step of screening involves removing duplicate articles. Therefore, publications in the form of review articles, proceedings or conference papers, book chapters, early access, books, editorial materials, letters, and notes were excluded. There was no timeline, and only articles written in English were considered to avoid confusion and problems with translation.

Figure 2 shows the eligibility process for article selection in this study. The process is important because the retrieved articles are manually selected to ensure that all of the articles remaining after the screening process meet the specified criteria. The titles, abstracts, and main content of selected articles were studied comprehensively to make sure they met the requirements to accomplish the objectives of the study.

*2.4. Quality Assessment*

Once the selected articles were finalized, quality assessment was performed. The articles were evaluated according to the quality criteria stated earlier in the protocol. After that, the articles were read and their quality was judged based on how well they answered the research questions [58]:

RQ1: The types of traditional ecological knowledge (TEK).
RQ2: The impacts of TEK on economic activities.
RQ3: The impacts of TEK on human health.

According to Devendran et al. [58], the selected articles were chosen and analyzed by manually reading full-text articles. The next step was to allocate weights for response to each research question according to the evaluation of its quality against the stated criteria. The weights were allocated as defined below:

- 0 if the article did not answer the research question or give any data or information about it.
- 0.5 if there were any data or information that could only partly answer a question but was still important.
- 1 if the data or information could answer the research question fully.

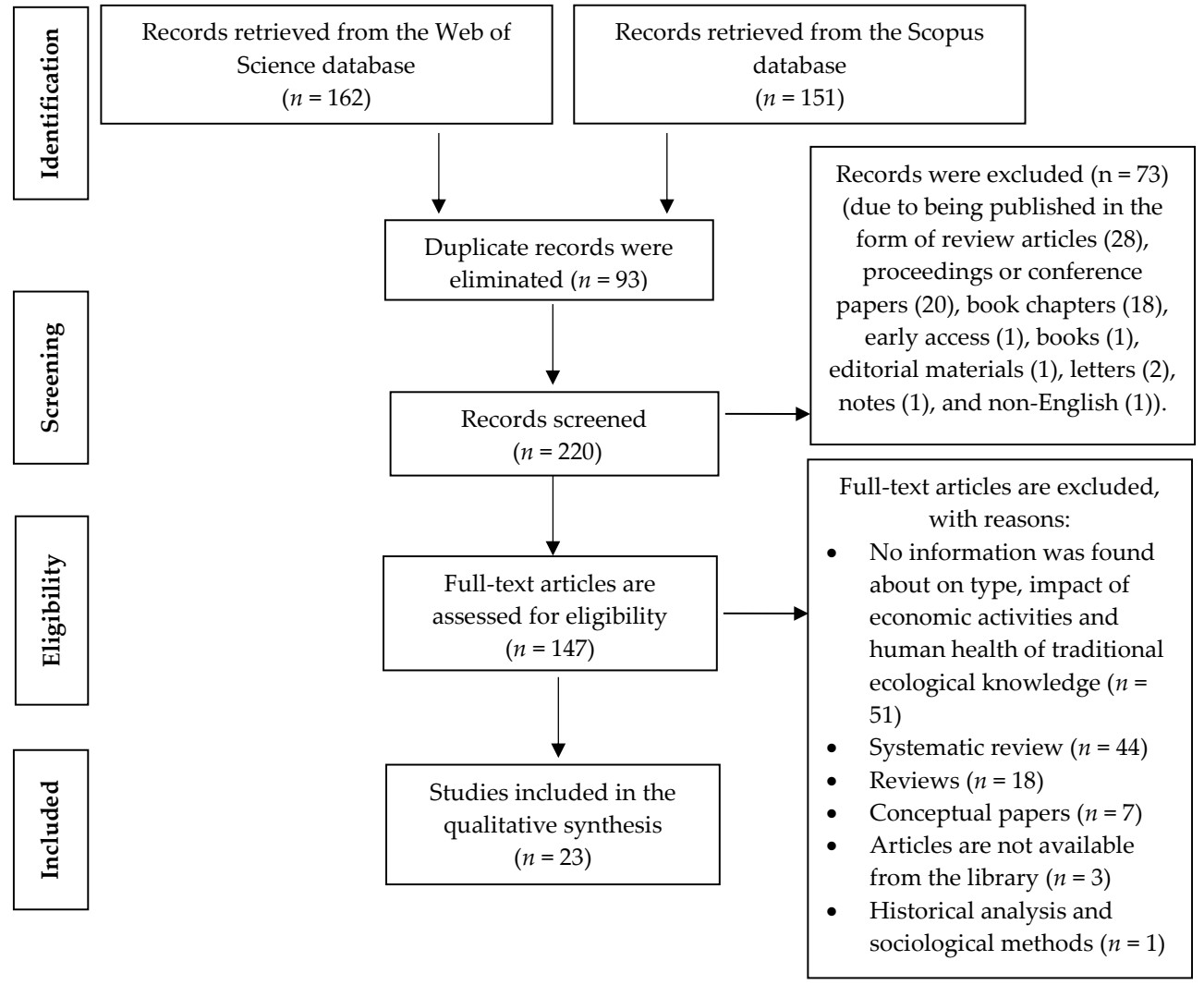

**Figure 2.** The flow diagram of the study (adapted from Moher et al. [53]).

*2.5. Data Abstraction and Analysis*

An integrated review has been used in the current study. Integrative review is a review approach that analyses and synthesizes diverse study designs, such as qualitative, quantitative, or a combination of the two, by turning one type into the other. The study used qualitative methodologies by thoroughly reviewing the selected papers' titles, abstracts, and contents. Data abstraction was performed in accordance with the research questions to ensure that the findings of the investigations could answer the specified research questions [3,52].

Following that, thematic analysis was conducted to establish the appropriate themes and sub-themes from the results. According to Braun and Clarke [59], thematic analysis is a method for describing, interpreting, and recording patterns in abstracted data. Compared to many other qualitative methodologies, thematic analysis has several advantages, including being easy to learn and implement, being accessible to and flexible for less experienced researchers, and allowing researchers to summarize and highlight essential points and interpret data from the results [59–62]. Furthermore, according to Boyatzis [63], thematic analysis allows researchers to relate information from diverse areas of a research issue in their own research.

## 3. Results and Discussion

The results from Scopus and WOS have been manually filtered using the defined keywords. Bibliographic and citation information was entered individually into Microsoft Word and Excel workbooks. The results of the bibliographic information included the name of the author(s), the title of the publication, the year of publication, the document type, and the country. Before making a final decision, a few measures must be taken. First and foremost, the article titles were assessed and the given criteria for inclusion and exclusion were used. Secondly, the papers were chosen by reading the abstracts: publications were discarded if they were unrelated to the current topic. Finally, the author manually evaluated the selected full-text publications.

As shown in Table 2, 147 articles remained after screening. The WOS database presented 115 articles pertaining to the keywords. The remaining 63 items were chosen based on their titles. Following that, selection based on the abstract left 48 articles. Finally, there were 19 publications on data interpretation. For Scopus, 32 articles were chosen based on keywords, of which 22 were chosen using a title-based filtering process. The abstracts of 11 publications led to their elimination. The remaining four articles were chosen for data interpretation.

**Table 2.** The searching process for the article.

| Databases | Total Result According to Stated Keywords | First Phase of Selection Filtered by Title | Second Phase Selection Filtered by Abstract | Final Result |
|---|---|---|---|---|
| WOS | 115 | 63 | 48 | 19 |
| Scopus | 32 | 22 | 11 | 4 |

The types of data source and methods of analysis of the selected studies are presented in Table 3. Table 4 displays the final article selection, which included 23 papers. The articles are arranged in the table from most to least recent. For this analysis, additional information included the nation where the study was conducted and was exclusively focused on research papers. Table 5 presents the publications that were chosen based on their year of publication. In addition, Figure 3 shows the impacts of traditional ecological knowledge on economic activities and human health.

**Table 3.** Type of data sources and method of analysis.

| Author(s) | Types of Data Sources | Method of Analysis |
|---|---|---|
| Abdullah et al. [7] | Interviews and questionnaire surveys | Analyzed qualitatively via the relative frequency of citation (RFC) using a random table method |
| Agarwal and Chandra [64] | Structured questionnaires along with field visits, group discussions, and key informant interviews | Analysis of a taxonomic group of plants |
| Hailemariam et al. [65] | Interviews, field observations, and group discussions | Using Pearson's chi-square test, direct matrix ranking, and pair-wise ranking |

**Table 3.** *Cont.*

| Author(s) | Types of Data Sources | Method of Analysis |
|---|---|---|
| Ali [8] | Field observations and interviews, focus group discussions, and a market survey | Interviewing 280 individuals (four border districts of Afghan Badakhshan, including Zebak, Ishkashim, Wakhan, and Shughnan, and four districts of Tajik Badakhshan, including Shughnan, Roshan, Ishkashim, and Wakhan) and asking them to name the medicinal plants they know and use. FGD consists of professionals and knowledgeable persons in the Ishkashim, Khorog, and Darwaz areas. A market survey was conducted to identify the most marketable and valuable MAP species for possible on-farm cultivation of certain species for local income generation, particularly for the women farmers |
| Shai et al. [9] | Interviews | Two ethnobotanical indices include frequency of citations (FC) and use value (UV) |
| Luo et al. [10] | Interviews, surveys, participatory rural appraisal, and focus group discussions | Quantitative indices, including the cultural food significance index (CFSI) and the relative frequency of citation (RFC) |
| Lakra [11] | Participant observations, interviews, and key informant discussions | Exploratory and descriptive study |
| Valle et al. [66] | Questionnaire surveys and passive observations | Using descriptive analysis |
| Manditsera et al. [67] | Interviews and questionnaire surveys | Using SPSS statistics for calculating the (relative) frequencies of consumption and a Chi-square test of independence |
| Atreya et al. [12] | Key informant interviews, focus group discussions, and a household survey | Using Binary Logistic Regression (BLR) |
| Kenny et al. [13] | Traditional Food Program | School-based setting, including the development of traditional knowledge and skills |
| Watts et al. [14] | Grey literature and reported catch/harvest statistics | Using Large Marine Ecosystem (LME) approaches |
| Bruschi et al. [15] | Interviews and group discussions | Using the Use Value (UV) index to quantify the relative importance of useful plants |
| Longvah et al. [16] | Focus group discussions | Descriptive and statistical analysis |
| Sujarwo et al. [17] | Direct observation, semi-structured interviews, key informant interviews, individual discussions, focus-group discussions, and questionnaires | Specimens were identified by the authors and other taxonomists and deposited in the herbarium of the Hortus Botanicus Baliense, Bali Botanical Garden |
| Salvi and Katewa [18] | Structured interviews, field observations, and group discussions | Determine the most commonly used plants, as well as the frequency and relative frequency of citations |
| Ao et al. [19] | Interview surveys | Identification of the collected mushrooms was done by standard microscopic methods and by studying the macroscopic and microscopic characters |
| Andriamparany et al. [68] | Interviews and questionnaire surveys | Two-step cluster analysis, discriminant analysis (DA), a one-way ANOVA (analysis of variance), and a generalised linear model (GLM) based on a Poisson distribution |

**Table 3.** *Cont.*

| Author(s) | Types of Data Sources | Method of Analysis |
|---|---|---|
| Towns et al. [20] | Interviews and focus group discussions | Participatory Action Research (PAR) methodology has been analyzed by inspection of the responses for recurring themes regarding the valuation of local foods |
| Uprety et al. [21] | Focus group discussions and key informant interviews | Using the Chi-square ($\chi2$) test of homogeneity, score based on clearly defined criteria and pairwise rank |
| Tiwari et al. [5] | Management practices were obtained from government records and through interviews with forest management officials, vegetation surveys, non-timber forest product (NTFP) utilization, and health care surveys | Participatory rural appraisal (PRA) |
| Kassam et al. [6] | Interview | Analytical approach |
| Singh et al. [22] | Explanatory research design | Qualitative approach |

**Table 4.** Details of selected papers after final selection.

| Author(s) | Title of Publication | Year | Document Type | Country |
|---|---|---|---|---|
| Abdullah et al. [7] | A Comprehensive Appraisal of the Wild Food Plants and Food System of Tribal Cultures in the Hindu Kush Mountain Range; a Way Forward for Balancing Human Nutrition and Food Security | 2021 | Article | Pakistan |
| Agarwal and Chandra [64] | Diversity and multipurpose utility of wild edible plant in Chopta—Mandal forest, Uttarakhand | 2021 | Article | India |
| Hailemariam et al. [65] | Ethnobotany of an indigenous tree *Piliostigma thonningii* (Schumach.) Milne-Redh. (Fabaceae) in the arid and semi-arid areas of South Omo Zone, southern Ethiopia | 2021 | Article | Ethiopia |
| Ali [8] | Effect and impact of indigenous knowledge on local biodiversity and social resilience in Pamir region of Tajik and Afghan Badakhshan | 2021 | Article | Pamir region of Tajik and Afghan Badakhshan |
| Shai et al. [9] | An Exploratory Study on the Diverse Uses and Benefits of Locally-Sourced Fruit Species in Three Villages of Mpumalanga Province, South Africa | 2020 | Article | South Africa |
| Luo et al. [10] | Diversity and use of medicinal plants for soup making in traditional diets of the Hakka in West Fujian, China | 2019 | Article | China |
| Lakra [11] | Sustainable Resource Management through Indigenous Knowledge and Practices—A Case of Food Security among the Baiga Tribe in India | 2019 | Article | India |
| Valle et al. [66] | Local knowledge for addressing food insecurity: The use of a goat meat drying technique in a rural famine context in Southern Africa | 2019 | Article | South Africa |
| Manditsera et al. [67] | Consumption patterns of edible insects in rural and urban areas of Zimbabwe: taste, nutritional value and availability are key elements for keeping the insect eating habit | 2018 | Article | Zimbabwe |

**Table 4.** *Cont.*

| Author(s) | Title of Publication | Year | Document Type | Country |
|---|---|---|---|---|
| Atreya et al. [12] | Factors Contributing to the Decline of Traditional Practices in Communities from the Gwallek-Kedar area, Kailash Sacred Landscape, Nepal | 2018 | Article | Nepal |
| Kenny et al. [13] | Linking health and the environment through education—A Traditional Food Program in Inuvik, Western Canadian Arctic | 2018 | Article | Canada |
| Watts et al. [14] | Inuit food security in Canada: arctic marine ethnoecology | 2017 | Article | Canada |
| Bruschi et al. [15] | Traditional knowledge on ethno-veterinary and fodder plants in South Angola: An ethnobotanic field survey in Mopane woodlands in Bibala, Namibe province | 2017 | Article | South Angola |
| Longvah et al. [16] | Mother and child nutrition among the Chakhesang tribe in the state of Nagaland, North-East India | 2017 | Article | India |
| Sujarwo et al. [17] | Traditional knowledge of wild and semi-wild edible plants used in Bali (Indonesia) to maintain biological and cultural diversity | 2016 | Article | Indonesia |
| Salvi and Katewa [18] | Documentation of folk knowledge on underutilized wild edible plants of Southern Rajasthan | 2016 | Article | India |
| Ao et al. [19] | Wild Edible Mushrooms of Nagaland, India: A Potential Food Resource | 2016 | Article | India |
| Andriamparany et al. [68] | Effects of socio-economic household characteristics on traditional knowledge and usage of wild yams and medicinal plants in the Mahafaly region of south-western Madagascar | 2014 | Article | Madagascar |
| Towns et al. [20] | Cultivated, caught, and collected: Defining culturally appropriate foods in Tallé, Niger | 2013 | Article | Nigeria |
| Uprety et al. [21] | Diversity of use and local knowledge of wild edible plant resources in Nepal | 2012 | Article | Nepal |
| Tiwari et al. [5] | Forest Management Practices of the Tribal People of Meghalaya, North-East India | 2010 | Article | India |
| Kassam et al. [6] | Medicinal Plant Use and Health Sovereignty: Findings from the Tajik and Afghan Pamirs | 2010 | Article | Tajik and Afghan Pamirs |
| Singh et al. [22] | Cultural significance and diversities of ethnic foods of Northeast India | 2007 | Article | India |

**Table 5.** Publications selected based on the year.

| Year | Author(s) |
|---|---|
| 2021 | Abdullah et al. [7]; Agarwal and Chandra [64]; Hailemariam et al. [65]; Ali [8] |
| 2020 | Shai et al. [9] |
| 2019 | Luo et al. [10]; Lakra [11]; Valle et al. [66] |
| 2018 | Manditsera et al. [67]; Atreya et al. [12]; Kenny et al. [13] |
| 2017 | Watts et al. [14]; Bruschi et al. [15]; Longvah et al. [16] |
| 2016 | Sujarwo et al. [17]; Salvi and Katewa [18]; Ao et al. [19] |
| 2015 | - |
| 2014 | Andriamparany et al. [68] |
| 2013 | Towns et al. [20] |
| 2012 | Uprety et al. [21] |
| 2011 | - |
| 2010 | Tiwari et al. [5]; Kassam et al. [6] |
| 2009 | - |
| 2008 | - |
| 2007 | Singh et al. [22] |

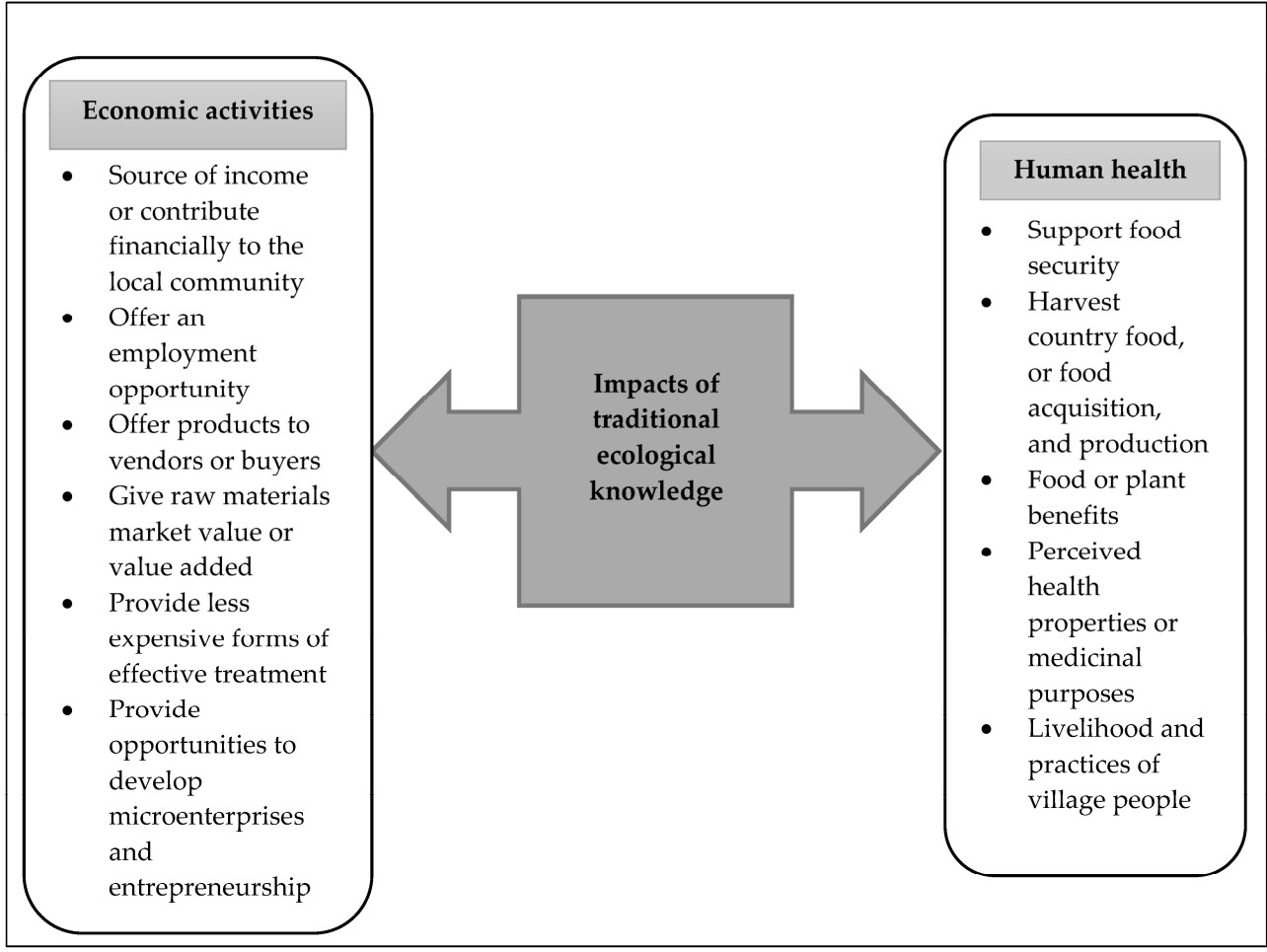

**Figure 3.** The impacts of traditional ecological knowledge on economic activities and human health.

*3.1. Overview of the Reviewed Articles*

3.1.1. Distribution by Country and Continent

Studies based in Asia are more frequent, the majority in India and Nepal, and there are fewer in Latin America and Europe. The results show that most studies come from developing countries such as India, Pakistan, Nepal, Indonesia, South Africa, and Bolivia. For instance, according to Abdullah et al. [7], Pakistan ranks sixth in terms of population and lower-middle-income status. It offers a huge diversity of natural resources, particularly plants, and all four seasons. If wild food plants are grown and used in a responsible way, they can be a valuable resource for underdeveloped communities and help a lot with hunger and malnutrition.

3.1.2. Categorization of Publication

Regarding the year of publication, four articles were published in 2021 [7,8,64,65], one article was published in 2020 [9], and three articles were published in 2019 [10,11,66]. Meanwhile, three articles were published in 2018 [12,13,67], three in 2017 [14–16], and three in 2016 [17–19]. Furthermore, one article was published in 2014 [68], one in 2013 [20], and one in 2012 [21]. There were another two articles published in 2010 [5,6], and lastly, one article in 2007 [22].

From what has been published over the years, it is clear that traditional ecological knowledge has become more and more important to research topics such as ethnobotany, food security, and local ways of making a living. Traditional knowledge of how to use wild edible plants and medicinal plants, however, is dwindling. Researchers recognize the importance of studying and documenting valuable information related to traditional knowledge on wild edible and medicinal plants because they provide essential nutrients and minerals, as well as economic potential for future generations.

*3.2. Types of Traditional Ecological Knowledge*

SLRs are invaluable tools to identify research trends and provide an opportunity for reflection within a field of study. In this systematic review, we set out to understand the impact of traditional ecological knowledge on indigenous people. Over time, traditional ecological knowledge has grown in diversity and scope, as has its audience, in many different nations around the world. Alongside this development, there has been an increase in the number of indigenous researchers committed to making sure that information is organized and presented in a way that is culturally beneficial.

The use of naturally sourced products, including parts of plants, insects, and animals, is well-studied and documented in the literature. A systematic review of prospective observational studies found that indigenous peoples are largely dependent on the natural environment to meet their daily livelihood needs, particularly on plant resources for food and medicines. Based on the plant parts used, the plant is classified into six categories, including flower, vegetable, fruits, leaves, seeds, and roots. A few studies have reported the large extent to which wild plants contribute to local food systems, including Carthamus oxyacantha, Pinus roxburghii seeds, and Marsilea quadrifolia in Pakistan [8]; Asparagus curillus, Fagopyrum dibotrys (D.Don) Hara, Myrica esculenta Ham. Steud., and Rubus spp., in India [64]; and Acacia rugata, Arisaema tortuosum, Artocarpus lakoocha, and Asparagus racemosus in Nepal [21]. Several studies reported that one of the most used parts of the plant is the fruit [69]. Strychnos madagascariensis, Berchemia discolor, Parinari capensis, Parinari curatellifolia, and Sclerocarya birrea are among the species locally consumed by indigenous peoples, as they have potential as alternative sources to meet dietary requirements and health needs, especially in rural communities.

In his widely reported and extensively explored literature on ethnobotany and indigenous types of knowledge, Fikret Berkes [70] proposes 11 categories of indigenous knowledge. TEK is often studied in the context of environmental and resource issues, but it is also relevant to many other disciplines, including ethnobotany and indigenous classification, resource harvesting and subsistence economy, resource use knowledge and

practice, social institutions for resource use, land use and occupancy, landscape knowledge and terminology, traditional knowledge education, oral history, indigenous ideology and worldview, decolonizing knowledge, and epistemology and knowledge systems [20].

Table 6 summarizes a selection of topics pertaining to local and traditional knowledge documented in the literature. It was discovered that the majority of TEK studied in the literature can be classified into four major classes. The categories undeniably overlap and are not intended to be exact. Documenting empirical species knowledge came first in traditional knowledge education, followed by investigation of ecological relationships and resource use systems. This knowledge also encompasses resource use practices and indigenous classification, which is further classified into wild edible plants, traditional practices and resource harvesting, and subsistence economics.

**Table 6.** Types of traditional ecological knowledge (TEK).

| Author(s) | Country | Types of Traditional Ecological Knowledge (TEK) |
| --- | --- | --- |
| Abdullah et al. [7] | Pakistan | Ethnobotany and indigenous classification |
| Agarwal and Chandra [64] | India | Resource harvesting and subsistence economy |
| Hailemariam et al. [65] | Ethiopia | Resource-use knowledge and practice |
| Ali [8] | Pamir region of Tajik and Afghan Badakhshan | Resource-use knowledge and practice |
| Shai et al. [9] | South Africa | Ethnobotany and indigenous classification |
| Luo et al. [10] | China | Ethnobotany and indigenous classification |
| Lakra [11] | India | Ethnobotany and indigenous classification |
| Valle et al. [66] | South Africa | Ethnobotany and indigenous classification |
| Manditsera et al. [67] | Zimbabwe | Ethnobotany and indigenous classification |
| Atreya et al. [12] | Nepal | Resource-use knowledge and practice |
| Kenny et al. [13] | Canada | Indigenous ideology and worldview |
| Watts et al. [14] | Canada | Resource-use knowledge and practice |
| Bruschi et al. [15] | South Angola | Ethnobotany and indigenous classification |
| Longvah et al. [16] | India | Resource-use knowledge and practice |
| Sujarwo et al. [17] | Indonesia | Ethnobotany and indigenous classification |
| Salvi and Katewa [18] | India | Ethnobotany and indigenous classification |
| Ao et al. [19] | India | Ethnobotany and indigenous classification |
| Andriamparany et al. [68] | Madagascar | Ethnobotany and indigenous classification |
| Towns et al. [20] | Niger | Resource-use knowledge and practice |
| Uprety et al. [21] | Nepal | Ethnobotany and indigenous classification |
| Tiwari et al. [5] | India | Resource-use knowledge and practice |
| Kassam et al. [6] | Tajik and Afghan Pamirs | Ethnobotany and indigenous classification |
| Singh et al. [22] | India | Traditional knowledge education |

### 3.3. The Impacts of Traditional Ecological Knowledge on Economic Activities

From Table 7, there are six sub-themes, namely, source of income or financial contribution to the local community [5,7–10,16–21,64,65,68]; offering an employment opportunity [5,9]; offering products to vendors or buyers [9,10,18,20,64,65]; providing raw materials at market value or value added [7,9,21,64]; providing less expensive forms of effective treatment [6]; and providing opportunities to develop microenterprises and entrepreneurship [19].

**Table 7.** The impacts of traditional ecological knowledge on economic activities.

| Themes | Sub-Themes |
| --- | --- |
| Source of income or contribute financially to the local community | (Abdullah et al. [7]; Agarwal and Chandra [64]; Hailemariam et al. [65]; Ali [8]; Shai et al. [9]; Luo et al. [10]; Longvah et al. [16]; Sujarwo et al. [17]; Salvi and Katewa [18]; Ao et al. [19]; Andriamparany et al. [68]; Towns et al. [20]; Uprety et al. [21]; Tiwari et al., 2010 [5]) |
| Offer an employment opportunity | (Shai et al. [9]; Tiwari et al. [5]) |
| Offer products to vendors or buyers | (Agarwal and Chandra [64]; Hailemariam et al. [65]; Shai et al. [9]; Luo et al. [10]; Salvi and Katewa [18]; Towns et al. [20]) |
| Provide raw materials market value or value added | (Abdullah et al. [7]; Agarwal and Chandra [64]; Shai et al. [9]; Uprety et al. [21]) |
| Provide less expensive forms of effective treatment | (Kassam et al. [6]) |
| Provide opportunities to develop microenterprises and entrepreneurship | (Ao et al. [19]) |

For the first sub-theme of source of income or financial contribution to the local community, the collection of forest products provides a means of supplementing household earnings. Forest products such as wild edible plants or medicinal plants are important in indigenous people's daily lives, and overuse of such resources threatens their livelihoods. Fruit species are another example of a forest product. Seasons such as the dry season and the start of the rainy season are crucial times for fruit species in Africa. To make additional money, the local populations make jams and juices from the native fruits throughout the ripening season [9]. Tiwari et al. [5] asserted that tribal communities use non-timber forest products (NTFPs) such as mushrooms, wild fruits, and vegetables that can be picked and sold in nearby markets.

The second sub-theme is offering employment opportunities. According to Tiwari et al. [5], for tribal people, forests provide a vital supply of food, fiber, freshwater, and building materials for subsistence, as well as being a source of economic income and a support system during difficult times. Additionally, the processing and sale of NTFPs can create more opportunities for the establishment of small-scale businesses at the local and regional levels, resulting in the creation of jobs for the community. Shai et al. [9] also claimed that picking fruit from farm-grown wild and semi-domesticated trees could increase employment in rural areas.

The third sub-theme is offering products to vendors or buyers. Local people produce goods from a variety of plant or fruit species—for example, wood from plants such as Lannea schweinfurthii, Diospyros mespiliformis, and Lannea edulis is used to produce wooden spoons, which are valuable kitchen utensils. Additionally, the local populations frequently use other species, including Pirinari capensis, Sclerocarya birrea, Vangueria infausta, and Strychnos madagascariensis, to offer a variety of beverages, including juice and wine, jam, oatmeal, and sweets. Recently, there has been a greater understanding of the value of indigenous plants for the development of new products, and many new items have been introduced [9]. Furthermore, Luo et al. [10] reported that the local community gathering edible plants and herb traders would frequently travel to rural areas to purchase medicinal herbs harvested in nature by villagers and then sell them at a profit in urban areas. The local communities were known for gathering edible plants. Most of the plants that were harvested were sold fresh, dried, or in simple packaging.

Providing market value or value added to raw materials is the fourth sub-theme. For example, the study by Abdullah et al. [7] stated that there are ten marketable plant species, including wild fruits and vegetables. Most species were available fresh in the market as well as dried during the off-season. Additionally, other kinds of wild food plants were used in herbal teas, decoctions, and fresh drinks. Meanwhile, Agarwal and Chandra [64] found that a few wild edible species were utilized to make, for instance, jam, chutney,

pickles, jelly, and squash, adding value to ensure high profits compared to selling the raw resources. The findings of the study on squash preparation included a cost-benefit analysis of using Rhododendron Arboreum squash. The production of squash boosted the price of Rhododendron Arboreum flowers from INR 40 to 70 per liter, with a net profit of 30 rupees per liter. Although the profit was significantly high, it was only available in certain research areas. It was discovered that locals can expect to earn an average of INR 15,000 per liter throughout the flowering season. It may be said that adding value to wild edible plants in the form of a byproduct is a profitable strategy that provides significant returns to people living in rural areas and enhances the longevity of various species. Shai et al. [9] addressed the diversity of indigenous knowledge and the continuing importance of fruit species. The majority of fruit species, which are frequently collected from the wild, are easiest to acquire in the summer. These fruit species can generally improve the population's wellbeing, the local economy, and the food production sector.

The next sub-theme is providing less expensive forms of effective treatment. Kassam et al. [6] claimed that despite periods of sociocultural and ecological shift, the local people of the Pamirs have preserved their understanding of a wide variety of remedial herbs. Medicinal plants support health sovereignty by offering significant healthcare possibilities. According to the findings of the study, 58 plants have 310 different practices in 63 categories of treatment and prevention, including analgesics, liver stimulants, dermatological stimulants, kidney stimulants, and hypotensives. Since nutrients are essential for maintaining human health, food is typically seen as medicine among the Pamirs, but not all medicines are considered food. Even though most food plants were categorized under medicinal plants, food and other remedial herbs can be classified by their usage and perceived benefits.

The last sub-theme of the impact of TEK on economic activities is providing opportunities to develop micro-enterprises and entrepreneurship. According to a study by Ao et al. [19], wild edible mushrooms (WEMs) are a widely demanded source of food in Nagaland. The locals use mushrooms to make soups, chutney, salads, and a variety of side dishes. Edible WEM types are in high demand during the growing season, and they can be purchased at local markets for between INR 50 and 250 per packet. Indigenous WEM expertise held by the locals offers major prospects for the growth of microbusinesses and entrepreneurship. Sustainability may be accomplished in this situation. WEMs help the economy a lot and make sure there is enough food for everyone.

### 3.4. The Impacts of TEK on Human Health

From the findings, the five sub-themes identified in Table 8 are supporting food security [19,65]; harvesting country food, food acquisition, and production [22]; food or plant benefits [7,9,11,13,14,16–21,64,66–68]; perceived health properties or medicinal purposes [5,6,8–10,12–15,17,21,22,64,65,67]; and livelihood and practices of village people [5–9,12,13,15–17,19–22,64–68]. A literature gap is found on the sub-theme of supporting food security [19,65] due to the current issue of food security under the sustainable development goals (SDGs). There is a further gap relating to the sub-theme of harvesting country food, food acquisition, and food production [22] because of the less popular traditional production process due to Industry Revolution 4.0 (IR 4.0) in developing countries such as India, Pakistan, and Malaysia.

**Table 8.** The impacts of traditional ecological knowledge on human health.

| Themes | Sub-Themes |
|---|---|
| Support food security | (Hailemariam et al. [65]; Ao et al. [19]) |
| Harvest country food, or food acquisition, and production | (Singh et al. [22]) |
| Food or plant benefits | (Abdullah et al. [7]; Agarwal and Chandra [64]; Shai et al. [9]; Lakra [11]; Valle et al. [66]; Manditsera et al. [67]; Kenny et al. [13]; Watts et al. [14]; Longvah et al. [16]; Sujarwo et al. [17]; Salvi and Katewa [18]; Ao et al. [19]; Andriamparany et al. [68]; Towns et al. [20]; Uprety et al. [21]) |
| Perceived health properties or medicinal purposes | (Agarwal and Chandra [64]; Hailemariam et al. [65]; Ali [8]; Shai et al. [9]; Luo et al. [10]; Manditsera et al. [67]; Atreya et al. [12]; Kenny et al. [13]; Watts et al. [14]; Bruschi et al. [15]; Sujarwo et al. [17]; Uprety et al. [21]; Tiwari et al. [5]; Kassam et al. [6]; Singh et al. [22]) |
| Livelihood and practices of village people | (Abdullah et al. [7]; Agarwal and Chandra [64]; Hailemariam et al. [65]; Ali [8]; Shai et al. [9]; Valle et al. [66]; Manditsera et al. [67]; Atreya et al. [12]; Kenny et al. [13]; Bruschi et al. [15]; Longvah et al. [16]; Sujarwo et al. [17]; Ao et al. [19]; Andriamparany et al. [68]; Towns et al. [20]; Uprety et al. [21]; Tiwari et al. [5]; Kassam et al. [6]; Singh et al. [22]) |

The first sub-theme of the impact of TEK on human health is support for food security. Both during normal times and during times of famine, the local communities relied mostly on wild edible plants and fruits as a source of food, especially among agropastoral communities amid drought and food shortages. When there is a food crisis, selling edible plants and fruits gives households the chance to make more money. The fruit ripens between December and February, depending on the amount of rainfall. When the brown pod cracks and the seeds fall away, the pulp surrounding the seeds is eaten raw as a snack or as emergency food, mostly by children [65]. Furthermore, Ao et al. [19] studied WEM, identifying their medicinal purposes and nutritional values. As well as being a further source of income, especially for local communities, WEM can contribute to food security.

The second sub-theme is harvesting country food, or food acquisition and production. The traditional foods that women in the northeastern region in India process and prepare are clearly relevant to their sociocultural, ecological, spiritual, and physical wellbeing. The processing and preparation of ethnic meals show tribal women's creativity and rich culinary heritage, as well as their gradual learning of how to preserve the environment and their way of life as a whole. Tribal women have created a variety of sustainable harvesting techniques, such as adaptive management techniques, for culturally significant species that are used as food sources. Various forms of communication are used by the various tribal communities in northeast India to pass on traditional knowledge linked to the processing and preparation of ethical foods from one generation to the next [22].

The third sub-theme is food or plant benefits. Most studies concluded that wild edible plants had a significant impact on the food system. Wild edible plants are the non-cultivated plant species that are gathered or harvested by local communities from their surrounding ecosystems for use in their food systems. In addition, local populations rely heavily on remedies from wild edible plants. For instance, wild edible plants provide a nutritious diet; can help prevent various diseases; are a rich source of beneficial ingredients, including essential fatty acids, vitamins, and complex carbohydrates; and can help treat malnutrition to a significant extent [7]. Meanwhile, according to Luo et al. [10], Hakka communities frequently use medicinal herbs as essential ingredients in traditional food, such as when cooking soup. Every meal in the Hakka traditional food system includes a medicinal soup to improve physical condition and long-term health.

The fourth sub-theme is perceived health properties or medicinal purposes. The results showed that most studies featured perceived health properties and medicinal purposes based on traditional knowledge. For instance, Piliostigma thonningii, which was identified



in the study by Hailemariam et al. [65], is one of the most notable indigenous multipurpose trees and serves a range of purposes, including producing human food and animal fodder as well as providing potential health benefits. Additionally, the many phytochemicals found in parts of the tree are extracted and utilized to treat a variety of illnesses, including fever, toothache, respiratory problems, and wound healing. In Tanzania, the juice from trees is used to treat illnesses such as stomachache, coughs, and snake bites, while tree parts such as tender leaves are chewed. Additionally, the roots are used to treat illnesses such as coughs, colds, and body aches, in addition to treating women's conditions such as extended menstruation, hemorrhage, and miscarriage.

Shai et al. [9] also mentioned that some native fruit trees are utilized in traditional medicine in different ways. There are various treatments for diseases, including snakebites, syphilis, and gonorrhea, as well as conditions including overly irritable bowel movements and sexually transmitted diseases. In addition, a variety of diseases and ailments—namely, TB, flu, toothache, gonorrhea, snakebites, infertility, headaches, and diarrhea—are commonly cured using fruit species. Traditional medical practitioners show that soaking and boiling plant parts in warm water for a certain amount of time is an effective way to treat the above condition.

The last sub-theme of the impact of TEK on human health is the livelihood or practice of village people. Based on the results, most studies explore the potential of wild edible plants, indigenous fruit trees, and medicinal plants for sustainable livelihoods in local communities. According to Hailemariam et al. [65], *Piliostigma thonningii* is a multipurpose tree that serves a variety of functions for the locals' way of life—for instance, as a source of food, for medicinal purposes, and for multipurpose materials such as shade, huts, chairs, fences, and timber. According to Ali [8], the indigenous people of the Pamir mountains have extensive knowledge of their surroundings and natural resources due to their close contact with nature and long-standing interactions with ecosystems for subsistence. They have continuously taken part in many various types of species conservation, especially when it comes to plant resources for food and medicinal treatments. Plants are used to make a variety of foods, including dry fruits, nuts, salads, soups, herbal tea, fruits preserved in syrup, jams, and spices. Medicinal plant species are also used to treat wounds, menorrhagia, and other illnesses, as well as the sense organs of animals.

As evidenced by the literature, ethnobotanical studies have been major topics of research. Ethnobotany is a branch of knowledge that evolved from documenting empirical knowledge about species to the study of ecological relationships and resource use systems. According to Tuxill and Nabhan [71], ethnobotany may serve as a promising approach and method for development. It is inspired by the fact that for a long time people have used plants for food, medicine, shelter, building products, and decoration [72]. A division of botany known as ethnobotany examines the interactions between science and both people and plants. Ethnobotanists can therefore act as a bridge between the two [73].

At the same time, ethnobotanical research has revealed that plant–human interactions have various opportunities to address certain of the world's present challenges, such as hunger, poverty, nutritional status, sustainable agriculture, and health care services [74,75]. In the same way, food shortages and lack of security are still big problems for most rural people, especially in developing countries [76–78].

Since the beginning of time, local communities have relied on a range of native plant species because they are easily accessible from their local surroundings. It is indisputable that native wild food plants, particularly in developing countries, can help ensure food security [79]. So, ethnobotanists have played an essential role in deciphering and documenting these plant–human interactions, as well as gaining knowledge through interviews and surveys [80]. As a result, ethnobotanists can assist in achieving several of the United Nations' SDGs including goals 1, 2, and 3, which are focused on eradicating poverty, eradicating hunger, and promoting better health and wellbeing, respectively [75]. Based on the findings, the five sub-themes support food security [19,65]. According to Food and Agriculture Organization (FAO) [74], the primary dimensions of food security include

physical access to a supply of food, affordability of food, consumption of food, and stability of the dimension over time. Locally grown plants can significantly reduce diet-related malnutrition because they are rich in vitamins and minerals [81,82]. Researchers from all around the world have already compiled a vast quantity of information on using wild plants as a food source. The results from a Google Scholar search using the keyword "wild food resources" show 1,740,000 results within the years 2000–2020. The findings indicate that a significant diversity of wild food plants consumed by various indigenous cultures has already been identified. By scientifically confirming the nutritional composition of wild and traditional food plants, they serve an essential role in supplementing nations' existing food baskets.

## 4. Conclusions

In conclusion, traditional ecological knowledge plays an essential role in indigenous peoples' economic and physical wellbeing. The current study systematically reviewed previous studies on the effects on economics and health of indigenous peoples' traditional ecological knowledge. This study employed an SLR approach, and 23 articles were reviewed for their quality. Based on the findings and analysis, two main themes emerged: economic activities, with six sub-themes, and health, with five sub-themes.

Most researchers have stated that in addition to offering food, nutrients, and medicinal benefits to the local communities, wild edible plants and medicinal plants have the potential to improve living standards for the people if they are managed sustainably. For instance, it is argued that it is crucial to explore which medicinal plants are traded in local markets, exported to regional and worldwide markets, and used for what purposes in the Pamirs. These multi-valued resources are in danger due to a number of anthropogenic and natural factors, including urbanization of the research site communities, land use change, habitat destruction, overharvesting, overgrazing, and harmful species. It is important to take care of these natural resources in a way that is good for local communities and keeps biodiversity safe.

This could also help to protect cultural and genetic diversity. Coordinated efforts between the local government and communities are needed to guarantee the preservation of crucial biodiversity knowledge that promotes both human and environmental wellbeing. Additionally, it exemplifies the potential for research, education, and community-based group collaborations in indigenous communities to address clear policy directions on the acknowledgment of traditional foods and associated knowledge systems, as well as key food security concerns. If not, it also helps to accelerate their extinction and devastates future generations' access to the therapeutic, economic, and dietary benefits that these plants and animals may offer.

The review has allowed us to identify some research gaps. For instance, in the Pamir mountain communities and Nagaland, there is wide knowledge of wild edible plants that are crucial for health and food security. However, there is little research on this issue. Furthermore, there is a lack of documentation of the local communities' traditional knowledge, particularly in relation to wild food plants, medicinal plants, and related knowledge and traditions. Additionally, it is crucial to research and record the ethnic foods that indigenous people eat in order to comprehend their eating habits, food availability, nutritional and therapeutic benefits, and connected cultural and social features. Another problem is that although locals depend on wild edible and medicinal plants for all of their nutritional needs, traditional knowledge of how to use these plants is rapidly disappearing. Due to this, there is a knowledge gap between the older and younger generations. In addition, certain wild edible plants have become endangered due to excessive harvesting. The sustainability of biological diversity is being seriously threatened by the diminishing indigenous knowledge, which has been identified as one of the key dangers. The associated traditional knowledge may disappear if no effort is made to educate the next generation about the significance of these plants.

This review makes several recommendations for future studies. The first recommendation is to develop policies that recognize the value of traditional knowledge and practices in the local adaptation plan while encouraging benefit sharing among stakeholders. We also propose that action is taken in local-level plans to address the decline of traditional knowledge and practices. Next, it is suggested that future applied research should concentrate on examining novel approaches to community empowerment and involvement, or on using community-based research techniques to better represent the needs of local communities and produce higher-quality research outcomes. In addition, public awareness, community-based management, biodiversity conversations, and cultivation should be fostered on all levels, and germplasm should be collected. The findings indicate that it is critical to cultivate the most commonly used food and medicinal plants, and to conduct additional research on the nutritional profiles and processing methods of all of the species reported in order to explore alternative sources of nutrition. The other recommendation is to increase scientists' responsibility significantly, beyond knowledge production and transfer, by facilitating a dialogue between different forms of knowledge to create synergy. This can be accomplished by determining how to improve knowledge co-production, bolstering neighborhood associations and their networks, updating agricultural educational facilities, and educating decision-makers. Additionally, the 2030 Agenda for Sustainable Development is distinctive in that it encourages action from all countries, including those with low, middle, and high incomes, in order to advance prosperity and safeguard the environment. Beyond governments, other parties will have to be engaged in the implementations, such as non-governmental organizations, indigenous peoples and local communities, women's groups, young people, and the business and financial community [83].

**Author Contributions:** Conceptualization, J.M.S., S.N.A., K.O., T.R.T.M. and N.A.S.; methodology, N.A.S.; validation, J.M.S., S.N.A. and N.A.S.; formal analysis, J.M.S., S.N.A. and N.A.S., writing—original draft preparation, N.A.S.; writing—review and editing, J.M.S., S.N.A., K.O., T.R.T.M. and N.A.S.; supervision, J.M.S. All authors have read and agreed to the published version of the manuscript.

**Funding:** This work was supported by the Ministry of Higher Education (MOHE) under the Fundamental Research Grant Scheme (FRGS), FRGS/1/2019/WAB13/UMT/02/1.

**Institutional Review Board Statement:** Not applicable.

**Informed Consent Statement:** Not applicable.

**Data Availability Statement:** All data generated or analyzed during this study are included in this publication.

**Acknowledgments:** The authors wish to express their appreciation to Universiti Malaysia Terengganu and Ministry of Higher Education Malaysia in financing this research under the Fundamental Research Grant Scheme (FRGS), Vote No. 59565.

**Conflicts of Interest:** This manuscript has not been published or presented elsewhere in part or entirety and is not under consideration by another journal. There are no conflict of interest to declare.

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
