# Peer review of "The Impacts of Traditional Ecological Knowledge towards Indigenous Peoples: A Systematic Literature Review"

_sustainability, doi:10.3390/su15010824_

Round 1
Reviewer 1 Report
This article has been written very well.
The writing flows is good, the point address is clear and no mistakes in term of the format. There are minor suggestion to improve the content of this manuscript.
1. In introduction, i wish to see a clearer map of indigenous people across the world. Please name a few of this group of people from different countries.
2. Please highlight which part of your studies related to sustainable development goal. Please specify it. How your studies match with the theme of this journal "sustainability". Adress this in introduction
3. In Table 8, you have listed all the themes that related to indigenous people. Why the first two theme (food security and harvest) was less popular as compared to other themes? Please justify.
Author Response
Response to Reviewer 1 Comments
Introduction
Point 1: In introduction, I wish to see a clearer map of indigenous people across the world. Please name a few of this group of people from different countries.
Response 1: We add a map of the names of tribes or communities in the areas of the selected studies in Figure 1.
Point 2: Please highlight which part of your studies related to sustainable development goal. Please specify it. How your studies match with the theme of this journal "sustainability". Address this in introduction
Response 2:
- SDG
This research makes a contribution to the sustainable development goals by addressing the objectives of goal 1, which is to end poverty; goal 2, which aims to end world hunger; and goal 3, which aims to ensure that all people enjoy good health and wellbeing. According to statistics gathered by the World Bank in 2016 [83] from 90 different countries, indigenous peoples make up 15 percent of the world's population of people who live in extreme poverty, making them the poorest of the poor. As a direct consequence of this, the second goal is to increase the productivity of indigenous peoples while simultaneously doubling their incomes. Goal 3 places an emphasis on wellbeing by aiming to achieve universal health coverage, encompassing financial risk protection, access to quality essential health care services, and access to essential medicines and vaccines that are safe, effective, of high quality, and affordable for all people.
- Sustainability
Indigenous peoples are not only the keepers of one-of-a-kind belief systems and knowledge systems, but they also possess invaluable knowledge regarding sustainable practices for the management of natural resources. This is because indigenous peoples have been living on their land for a long time. They have a special connection to the land that has been in their family for many years, and they make productive use of it. In terms of both their current level of physical wellbeing and the cultural customs they have passed down through the generations, the land that their ancestors once inhabited is of the ut-most significance to their continued existence as a distinct nation. Indigenous peoples have their own distinct concept of development that is based on the values, visions, needs, and priorities that have been ingrained in their culture for generations. As a result, it is critical to ensure that their traditional knowledge is preserved for future generations, par-ticularly when it comes to the use of various medicinal plants.
Results and Discussions
Point 3: In Table 8, you have listed all the themes that related to indigenous people. Why the first two theme (food security and harvest) was less popular as compared to other themes? Please justify.
Response 3: From the findings, the five sub-themes identified in Table 8 are supporting food security [48, 62]; harvesting country food, food acquisition, and production [66]; food or plant benefits [46, 47, 50, 52, 53, 54, 56, 57, 59, 60, 61, 62, 63, 64, 65]; perceived health properties or medicinal purposes [47, 48, 49, 50, 51, 54, 55, 56, 57, 58, 60, 65, 11, 20, 66]; and livelihood and practices of village people [46, 47, 48, 49, 50, 53, 54, 55, 56, 58, 59, 60, 62, 63, 64, 65, 11, 20, 66]. A literature gap is found on the sub-theme of supporting food security [48, 62] due to the current issue of food security under the sustainable development goals (SDGs). There is a further gap relating to the sub-theme of harvesting country food, food acquisition, and food production [66] because of the less popular traditional production process due to Industry Revolution 4.0 (IR 4.0) in developing countries such as India, Pakistan, and Malaysia.

Reviewer 2 Report
The authors worked to process a large number of articles using quantitative methods, as well as a significant number of articles using qualitative methods. The formal requirements for writing a journal article are met, step by step. The methodology is presented in detail and correctly. Bibliographic references are pertinent and comprehensive. And yet, the scientific contribution of this article is low, the descriptive character predominating.
I have no suggestions to give the authors, because they have done well what they set out to do, except perhaps for an attempt at innovative scientific interpretation, starting from the results of the analysis. Such an interpretation does not exist.
Author Response
Response to Reviewer 2 Comments
Results and Discussion
- Scientific approach
Point 1: The authors worked to process a large number of articles using quantitative methods, as well as a significant number of articles using qualitative methods. The formal requirements for writing a journal article are met, step by step. The methodology is presented in detail and correctly. Bibliographic references are pertinent and comprehensive. And yet, the scientific contribution of this article is low, the descriptive character predominating.
Response 1: The use of naturally sourced products, including parts of plants, insects, and ani-mals, is well studied and documented in the literature. A systematic review of prospective observational studies found that indigenous peoples are largely dependent on the natural environment to meet their daily livelihood needs, particularly on plant resources for food and medicines. Based on the plant parts used, the plant is classified into six categories, including flower, vegetable, fruits, leaves, seeds, and roots. A few studies have reported the large extent to which wild plants contribute to local food systems, including Carthamus oxyacantha, Pinus roxburghii seeds, and Marsilea quadrifolia in Pakistan [49]; Asparagus curillus, Fagopyrum dibotrys (D.Don) Hara, Myrica esculenta Ham. Steud., and Rubus spp., in India [47]; and Acacia rugata, Arisaema tortuosum, Artocarpus lakoocha, and Asparagus racemosus in Nepal [65]. Several studies reported that one of the most used parts of the plant is the fruit [81]. Strychnos madagascariensis, Berchemia discolor, Parinari capensis, Parinari curatellifolia, and Sclerocarya birrea are among the species locally consumed by indigenous peoples, as they have potential as alternative sources to meet dietary requirements and health needs, especially in rural communities.
